# Parkinson’s Disease-Related Brain Metabolic Pattern Is Expressed in Schizophrenia Patients during Neuroleptic Drug-Induced Parkinsonism

**DOI:** 10.3390/diagnostics13010074

**Published:** 2022-12-27

**Authors:** Ivan Kotomin, Alexander Korotkov, Irina Solnyshkina, Mikhail Didur, Denis Cherednichenko, Maxim Kireev

**Affiliations:** 1N.P. Bechtereva Institute of the Human Brain, Russian Academy of Sciences, Academika Pavlova Street 9, Saint Petersburg 197376, Russia; 2Institute for Cognitive Studies, Saint Petersburg State University, Saint Petersburg 197376, Russia

**Keywords:** drug-induced parkinsonism, schizophrenia, neuroleptic treatment, Parkinson’s disease-related pattern, 18F FDG PET

## Abstract

Drug-induced parkinsonism (DIP) is a frequent parkinsonian syndrome that appears as a result of pharmacotherapy for the management of psychosis. It could substantially hamper treatment and therefore its diagnosis has a direct influence on treatment effectiveness. Although of such high importance, there is a lack of systematic research for developing neuroimaging-based criteria for DIP diagnostics for such patients. Therefore, the current study was aimed at applying a metabolic brain imaging approach using the 18F-FDG positron emission tomography and spatial covariance analysis to reveal possible candidates for DIP markers. As a result, we demonstrated, to our knowledge, the first attempt at the application of the Parkinson’s Disease-Related Pattern (PDRP) as a metabolic signature of parkinsonism for the assessment of PDRP expression for schizophrenia patients with DIP. As a result, we observed significant differences in PDRP expression between the control group and the groups with PD and DIP patients. Similar differences in PDRP expression were also found when the non-DIP schizophrenia patients were compared with the PD group. Therefore, our findings made it possible to conclude that PDRP is a promising tool for the development of clinically relevant criteria for the estimation of the risk of developing DIP.

## 1. Introduction

Parkinsonism is a clinical syndrome that includes bradykinesia, rigidity, and tremors. The most common cause of parkinsonism is Parkinson’s disease (PD), while the second most common etiology is drug-induced parkinsonism (DIP) [1,2,3]. DIP can be induced by typical and atypical antipsychotics, antiepileptic medications, and calcium channel blockers [3]. It was shown in about 80% of patient’s treatment with typical antipsychotics such as haloperidol, promazine, chlorpromazine, perphenazine, pimozide and fluphenazine that these drugs are associated with more than one kind of extrapyramidal symptom [4]. Atypical antipsychotics (risperidone, clozapine, quetiapine, olanzapine and aripiprazole) are associated with a lower rate of DIP, see [5].

The DIP associated with antipsychotic therapy is of particular clinical importance. The effectiveness of pharmacotherapy for numerous psychiatric conditions, including schizophrenia and bipolar disorders, could be significantly hampered by the appearance and aggravation of parkinsonian symptoms. It plays a crucial role in the treatment of psychosis when therapy requires achieving a balance between reducing extrapyramidal symptoms and avoiding recurrent psychosis. Moreover, in some cases, the reduction in the dosage of antipsychotic pharmacotherapy or its disruption could lead to the aggravation of extrapyramidal symptoms [6,7].

The cause of the symptoms of parkinsonism is recognized to be pathological changes in the neural activity of several interconnected brain structures. However, the neural mechanisms underlying DIP remain underinvestigated and any criteria for the estimation of the risk of developing DIP are lacking. To fill this gap, we conducted the current observational study aimed at revealing pathological brain reorganization in DIP during antipsychotic treatment of schizophrenia and PD using 18F-FDG positron emission tomography (PET). For these purposes, we used the approach based on the evaluation of the spatial covariance analysis of the 18F-FDG distribution, the so-called Parkinson’s disease-related pattern (PDRP), which has been widely used in recent years [8,9,10]. PDRP is the commonly used method of PET diagnostics when motor symptoms are observed in patients with PD, which has been repeatedly validated [11,12] and allows an objective assessment of disease activity in individual subjects [13]. To our knowledge, the expression of PDRP in patients with DIP has not been studied before. We hypothesize that, depending on the observation of DIP symptoms during antipsychotic therapy for schizophrenia, PDRP may be differentially expressed and serve as a candidate for a clinically relevant prognostic and diagnostic biomarker for DIP.

## 2. Materials and Methods

### 2.1. Study Population

#### 2.1.1. Schizophrenia Patients with and without DIP

Retrospective analysis of clinical records (Psychiatry Department of the Clinic of IHB RAS, Dr. A.N. Chomsky) was used for patient selection. The inclusion criteria were: schizophrenia diagnosis (ICD-10 code F.20.0) applied by the psychiatrist at hospitalization, presence and availability of PET/CT and MRI data, neurological stage records, health complaints records and medical prescription sheet in the hospitalization period, especially two days before and two days after the FDG-PET investigation. Exclusion criteria were age younger than 18 years, focal lesions of the brain structure detected by magnetic resonance imaging, another PET scanner model (see below), and artifacts of the PET image.

The final study group consisted of 28 patients with schizophrenia hospitalized and undergoing the FDG-PET study between 2017 and 2019. Clinical and demographic characteristics are given in Table 1. Fourteen patients were assigned to the group without extrapyramidal symptoms (non-DIP group) and fourteen patients were assigned to the group with extrapyramidal symptoms (DIP group). Chlorpromazine equivalent of antipsychotic doses was calculated using minimum effective dose method ratios [14] excluding amisulpride, chlorprothixene and zuclopenthixol, which were assessed by international consensus ratios [15]. Detailed psychopharmacological anamnesis during the period of FDG-PET investigation is presented in Appendix A, Table A1.

#### 2.1.2. Patients with Parkinson’s Disease

In total, 19 patients with Parkinson’s disease were studied, among them, nine with rigid akinetic, eight with tremor-dominated and two with mixed subtypes were selected for the derivation of the PDRP (PD group). Clinical and image data from another 13 rigid akinetic and two tremor-dominated PD patients were used for pattern validation (PDv group). All patients underwent FDG-PET between 2015 and 2019. The medical records were reviewed to verify the status of the disease and obtain clinical information.

### 2.2. Healthy Subjects

Since Russian radiation safety legislation limits the allowed doses for non-medical sessions to 1 mSv per year, a control group of healthy volunteers scanned on the same PET system and with the same settings was not available. Therefore, FDG-PET data of 19 oncologic patients (excluding neurooncology and metastatic tumors) and microvascular angina (MA) patients without neurologic and psychiatric disorders and medication scanned between 2016 and 2020 were used as control group (C). In addition, a healthy control group comprising 18 FDG-PET scans from the AIMN dataset [16] was used for validation purposes.

All clinical and demography data of PDRP derivation and validation sets are shown in Appendix A, Table A2.

### 2.3. FDG-PET Scanning

FDG-PET scans of patients from the Schizophrenia, PD and C groups were performed as a clinical routine procedure at IHB RAS between 2015 and 2020. Radiopharmaceutical was prepared on the site with a GE PET trace 800 cyclotron and in-home developed automated module [17]. Patients scanned on GE Discovery 710 system with 16-slice CT. After intravenous administration of 3–6 mCi of FDG, according to the patient’s body weight, patients lay for 40–50 min in a darkened room, followed by low-dose CT of the brain and 6-min brain emission scan in 3D time-of-flight detection mode. Ordered subset expectation maximization protocol with point spread function, 36 subsets and 2 iterations were used for reconstruction. In patients with PD and schizophrenia, as well as MA and oncological diseases, there were no special recommendations to limit medication intake on the day of PET examination. Fasting was prescribed for 6–8 h before the scan C group. Subjects from AIMN dataset were scanned in 3D mode on the GE Discovery ST-E system with closed eyes and the OSEM reconstruction protocol but more precise details are unavailable. Patients who scanned in other PET systems were excluded from the study.

### 2.4. Image Data Preprocessing

FDG-PET scans were visually checked for image artifacts. FDG-PET scans were visually checked for image artifacts. Further PET, low-dose CT, and, if available, MRI series visually assessed for rough focal structural or ametabolic lesions. DICOM PET series was then converted to Nifti format using Mricron [18], spatially normalized using SPM12 toolbox Oldnorm batch [19] with [18F]-FDG PET dementia-specific template [20] and smoothed with 12 mm Gaussian kernel. Finally, images were converted from Nifty to Analyze format for further SSM-PCA analysis using ScanVP ver. 7.0 toolbox [21]. All toolboxes were launched under the Matlab 2014b environment for Windows.

### 2.5. PDRP Derivation and Validation

PDRP was obtained using the method described elsewhere [22]. The preprocessed FDG-PET images of the PD derivation set and the control group were included in SSM-PCA procedure. The signals from white matter and cerebrospinal fluid were removed by applying threshold of 0.35 from mean subject brain activity. Activity values of remaining image voxels were then logarithmically transformed and the subject-to-voxel matrix of whole dataset was built. Global centering of data was performed by removing column and row mean values. The set of residual subject profiles were then decomposed into the principal components (PCs) and the subject scores matrix during PCA. Components that explained the first 50% of the total variance were included in further analysis. Student’s *t*-test (*p* < 0.05) on subject scores of each of these PCs was used for component selection. PCs that showed significant difference between PD and C scores were selected and prospective score evaluation procedure of subjects from PD validation set (PDv) and AIMN healthy controls was performed.

PCs that have shown significant differences during the pairwise cross-validation procedure in which the PD groups were compared with control groups (*p* < 0.05) were linearly combined. Logistic regression on subject scores was used to obtain weighting coefficients and the resulting combination was defined as PDRP. PDRP voxels were overlaid on T1 MRI template for visualization.

### 2.6. PDRP Evaluation in Schizophrenia

Preprocessed PET images of schizophrenic patients with and without extrapyramidal symptoms were scaled using the prospective score evaluation procedure relative to PDRP. The values obtained for these groups were compared with each other and with PD and C groups.

### 2.7. Statistical Analysis

The normal distribution of subject scores in the comparison groups was assessed by Lilliefors test (*p* > 0.05). The equality of variances was tested using Levene’s criterion (*p* > 0.05). One-way ANOVA (*p* < 0.05) and Bonferroni–Holm procedure for multiple comparisons correction (FWER < 0.05) were performed for statistical testing of hypotheses about equality of PDRP scores between study groups. In cases of violation of normal distribution or equality of variances, nonparametric analogs of ANOVA were used. Spearman analysis was used to test correlations (*p* < 0.05). The calculations were performed in StatSoft Statistica Software ver. 12 and with in-home Python script.

## 3. Results

C, PD and Schizophrenia groups differed significantly in age (ANOVA post-hoc *p* < 0.05); however, there were no differences between non-DIP and DIP groups. There was no statistical evidence that PANSS scores, as well as antipsychotic equivalent doses, differed between non-DIP and DIP patients.

In the DIP group, six patients received at least one antipsychotic at a high dose (four of them received a typical one) versus only two patients in the non-DIP group (one of them received a typical one). At the same time, non of the patients in the DIP group received only one neuroleptic at a low dose versus two such patients in the non-DIP group.

After PCA, six principal components, which explained 50% of the variance, were obtained. Subject scores of PC1 and PC3 (16% and 8% of variances explained) significantly differed between the C group and the PD and PDv sets, as well as between the AIMN controls and both the PD groups. The linear combination of these two components was defined as PDRP. The distribution of the Z-transformed positive and negative voxel weights of the pattern is displayed in Figure 1.

There was no significant sex (*p* > 0.05) and age (R = 0.08, *p* > 0.05) dependencies of PDRP expression. Comparison of the PDRP scores in the C, non-DIP, DIP and PD demonstrated an uptrend (ANOVA *p* < 0.05, Figure 2). Finally, we did not obtain a significant correlation between PDRP expression and the Chlorpromazine equivalent of administered antipsychotic medications (R = 0.14, *p* > 0.05).

## 4. Discussion

In our study, we observed a statistically significant influence of disease factors on PDRP expression. These results are partially in line with our expectations and confirm that the expression of PDRP can be considered as a potential brain metabolic signature of medication-induced parkinsonism. However, the results also suggest that abnormal PDRP expression is observed in the group of patients receiving antipsychotics, even in the absence of clinical manifestations of DIP. It should be noted that the frequentists statistic we used did not allow us to conclude about the similarity in expression of PDRP when the result is non-significant, see also [23]. As follows from the graph (Figure 2), the mean Z-score for the non-DIP group has an intermediate position between the control group and the DIP group. However, taken together, our results allow us to speculate that the expression of PDRP and the metabolic signature of parkinsonism gradually manifests itself in a joint analysis of all four groups.

Patients in both groups received both typical (four different drugs) and atypical (eight different drugs) neuroleptics in low, therapeutic, and high doses. Moreover, in both groups, typical neuroleptics received the same number of patients. Due to the heterogeneity of pharmacological data, we used the Chlorpromazine equivalents to compare medication doses. Although the number of patients at high and low doses differed between groups and the median equivalent dose in the DIP group was higher than in the non-DIP, this difference was not statistically significant. Furthermore, there was no correlation between equivalent doses and PDRP scores. Thus, our data do not allow us to find differences between DIP and non-DIP groups and conclude about the effects of the type of antipsychotics or their dose on the expression of PDRP.

The functional neuroanatomy of PDRP involves relative glucose hypermetabolism in the thalamus, putamen/pallidum, pons, cerebellum, and motor cortex, as well as hypometabolism in the posterior parietal, occipital, and frontal cortices [12,24]. This reflects a complex reorganization of the brain that underlies the motor symptoms of parkinsonism.

The etiology and pathophysiological mechanisms of Parkinson’s disease have been extensively studied and, in most cases, can be determined in an individual patient. The etiology and pathophysiological mechanisms of DIP are less well known and can be considered from two predispositions. The first suggests a direct toxic effect of the drugs responsible for neurons by inhibiting mitochondrial respiratory function and contributing to irreversible cell death or pathway deficits in the dopaminergic structures of the area of nigrostriatal dopamine. The second takes into account the presence of a significant loss of dopaminergic neurons or/and the disturbance of connections between brain structures, observed in psychiatric patients with symptoms of DIP, which manifests itself under the toxic effects of neuroleptics. The latter idea finds a number of confirmations. Recently, after four years of follow-up, Jeong et al. [25] observed that DIP is closely associated with an increased risk of developing idiopathic Parkinson’s disease compared to the control group (patients with Diabetes Mellitus). The authors conclude that this association could be related not to the direct toxic effects of medication that led to irreversible neuronal damage, but rather to unmasking pre-existing subclinical idiopathic Parkinson’s disease. This view is also confirmed in clinicopathological studies, which showed that patients with DIP had pathological findings compatible with those underlying idiopathic PD [26,27]. Furthermore, it should be noted that disease that requires the administration of antipsychotics can be accompanied by motor disorders indicating the pathology of dopaminergic structures [28,29].

From this point of view, the results of a current study indicate that analysis of the expression of the Parkinson’s disease-related brain metabolic pattern may be useful in assessing the risk of developing drug-induced parkinsonism in patients receiving antipsychotic therapy. Although the results obtained are preliminary, the findings deserve further investigation and must be confirmed in a larger and independent cohort of patients receiving antipsychotic pharmacotherapy.

## Figures and Tables

**Figure 1 diagnostics-13-00074-f001:**
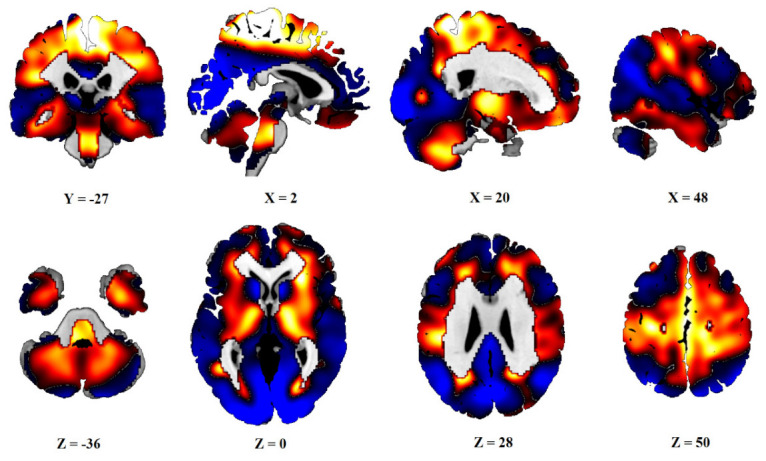
Topography of PDRP. Distribution of positive (red) and negative (blue) voxels of PDRP. Voxel values PDRP are overlaid on a T1 MRI template. Coordinates in the frontal (Y), sagittal (X) and axial (Z) planes are in Montreal Neurological Institute (MNI) standard space. The left side is on the left.

**Figure 2 diagnostics-13-00074-f002:**
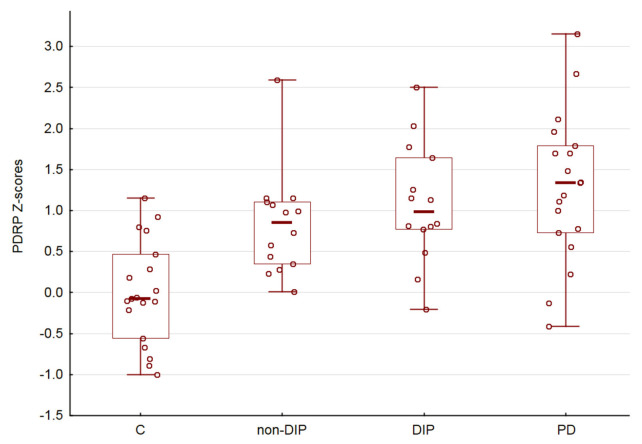
PDRP subject scores. Horizontal lines are medians, boxes reflect interquartile range, while whiskers—overall range.

**Table 1 diagnostics-13-00074-t001:** Demographic and clinical characteristics of schizophrenic patients with and without DIP.

Characteristics	DIP Group (N = 14)	Non-DIP Group (N = 14)	Mann– Whitney U
Sex, M/F	8/6	8/6	-
Age, median (IQR ^1^)	28 (23–33)	27 (24–39)	*p* > 0.05
PANSS ^2^ total, median (IQR)	88.5 (82–98)	91 (79–104)	*p* > 0.05
PANSS positive, median (IQR)	23 (19–27)	21 (18–23)	*p* > 0.05
PANSS negative, median (IQR)	23 (20–28)	23 (20–29)	*p* > 0.05
PANSS general, median (IQR)	44 (42–50)	47 (40–52)	*p* > 0.05
Chlorpromazine equivalent, median (IQR)	744 (375–1250)	517 (250–676)	*p* > 0.05

^1^ IQR, Interquartile range, ^2^ PANSS, Positive and Negative Syndrome Scale.

## Data Availability

The data analyzed in the current study are available from the corresponding author upon reasonable request.

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
