# Peer review of "Parkinson’s Disease-Related Brain Metabolic Pattern Is Expressed in Schizophrenia Patients during Neuroleptic Drug-Induced Parkinsonism"

_diagnostics, 2022, doi:10.3390/diagnostics13010074_

Round 1

Reviewer 1 Report

The report produced by authors quite interesting but I have some comments that need to answer.

1. Section 2.1.1 Schizophrenia patients with and without DIP: The DIP group i.e 14 schizophrenia patients might not got the same type of antipsychotic medicines. Some patients could have been treated with sedatives and hypnotics, anticonvulsant carbamazepine or other medicines in combination. In all cases parkinsonism syndromes may not be possible.How can you differentiate all these concerns?

2. The sentences 119-123 difficult to understand, needs correction.

3. The types of neuroleptics causing parkinsonism syndrome need to be discussed with reference.

4. Line 39-41:   Reference required.

5. Wheather administration of neuroleptic  with High dose, low dose and therapeutic dose causing parkinsonism, need to be clarified.

6. Name of the Hospital with physician details need to be highlighted to enlighten the report.

Author Response

We are very grateful to reviewers for valuable comments. Please see attechment for the point by point response.

Reviewer 2 Report

In general, this is a very interesting study and has certain practical significance. It can be used to assess the risk of drug-induced Parkinson's disease in patients receiving antipsychotic drugs. However, there are some errors in the article, which need to be carefully checked.

1. Line 12:“Drug-induced parkinsonism (DIP) is frequent parkinsonian syndrome” should be corrected to “Drug-induced parkinsonism (DIP) is a frequent parkinsonian syndrome”.

2. Line 13: “It could substantially hampers the treatment and its diagnostics” should be corrected to “It could substantially hamper the treatment and its diagnostics”.

3. Line 19: “to the our knowledge” should be corrected to “to our knowledge”.

4. Line 90: “All clinical and demography data of PDRP derivation and validation sets is shown at Appendix A, Table A1.” should be corrected to “are”.

5. Line 149: “differences” should be corrected to “difference”.

Author Response

(The authors gave the same response as above.)

Round 2

Reviewer 2 Report

It can be accepted in the current version.